# PRR adjuvants restrain high stability peptides presentation on APCs

**Bin Li[1]\*[†], Jin Zhang[1][†], Taojun He[1], Hanmei Yuan[1], Hui Wu[1], Peng Wang[2]\*, Chao Wu[1]\***

[1]Department of Laboratory Medicine, The Eighth Affiliated Hospital of Sun Yat-sen University, Shenzhen, China; [2]Department of Orthopedics, The Eighth Affiliated Hospital of Sun Yat-sen University, Shenzhen, China

## eLife Assessment

This **important** study provides interesting insights into the mechanisms of action of adjuvants. It shows that adjuvants, MPLA and CpG especially, modulate the peptide repertoires presented on the surface of antigen presenting cells, and surprisingly, adjuvant favored the presentation of low-stability peptides rather than high-stability peptides by antigen presenting cells. As a result, the low stability peptide presented in adjuvant groups elicits T cell response effectively. Evidence in support of these conclusions is **solid**, and this paper would be of interest to vaccinologists and immunologists.

**\*For correspondence:**
libin2005017@163.com (BL);
wangp57@mail.sysu.edu.cn (PW);
wuch57@mail.sysu.edu.cn (CW)

[†]These authors contributed equally to this work

**Competing interest:** The authors declare that no competing interests exist.

**Abstract** Adjuvants can affect APCs function and boost adaptive immune responses post-vaccination. However, whether they modulate the specificity of immune responses, particularly immunodominant epitope responses, and the mechanisms of regulating antigen processing and presentation remain poorly defined. Here, using overlapping synthetic peptides, we screened the dominant epitopes of Th1 responses in mice post-vaccination with different adjuvants and found that the adjuvants altered the antigen-specific CD4[+] T-cell immunodominant epitope hierarchy. MHC-II immunopeptidomes demonstrated that the peptide repertoires presented by APCs were significantly altered by the adjuvants. Unexpectedly, no novel peptide presentation was detected after adjuvant treatment, whereas peptides with high binding stability for MHC-II presented in the control group were missing after adjuvant stimulation, particularly in the MPLA- and CpG-stimulated groups. The low-stability peptide present in the adjuvant groups effectively elicited robust T-cell responses and formed immune memory. Collectively, our results suggest that adjuvants (MPLA and CpG) inhibit high-stability peptide presentation instead of revealing cryptic epitopes, which may alter the specificity of CD4[+] T-cell-dominant epitope responses. The capacity of adjuvants to modify peptide–MHC (pMHC) stability and antigen-specific T-cell immunodominant epitope responses has fundamental implications for the selection of suitable adjuvants in the vaccine design process and epitope vaccine development.

## Introduction

Adjuvants are an important composition of vaccines that enhance or prolong the immune responses to co-administered antigens post-vaccination. The most commonly used adjuvants stimulate antibody responses by relying on B-cells (*Levast et al., 2014*; *Petrovsky, 2015*). B-cell priming and memory forming are dependent on the cognate help of antigen-specific T-cells. T-cells also have direct effector functions against pathogenic infections. Thus, antigen-specific T-cells play a vital role in shaping the

immune response during vaccination. However, the mechanisms by which adjuvants regulate the specificity of antigen-specific T-cell responses remain poorly understood.

Antigen-specific T-cell responses mainly focus on several peptides derived from antigen amino acids called immunodominant epitopes (*Sant et al., 2005*). Although specific T-cell responses are provoked by the same antigen, their response epitopes may differ, leading to diverse effects. Peptides presented by antigen-presenting cells (APCs) provide the first major signal for T-cells priming and determining their specificity. Although adjuvants can activate APCs and increase the expression of co-stimulatory molecules of APCs (*Pulendran, 2004*), the effect of adjuvants on the peptide repertoires presented by APCs or modulation of epitope-specific T-cell responses remains unknown.

*Helicobacter pylori* infects more than half the global population and causes chronic gastritis, peptic ulcers, gastric mucosa-associated lymphoid tissue lymphoma (MALT), and gastric cancer (*Malfertheiner et al., 2023*; *Moss et al., 2024*). Studies in human and mouse models showed that the Th1 response provides more important protection against *H. pylori* than a humoral immune response. Unfortunately, no effective vaccines have been developed to drive T-cell responses (*Friedrich and Gerhard, 2023*). *H. pylori* provides an ideal experimental model to determine the mechanisms by which adjuvants regulate T-cell responses.

Studies on widely used adjuvants have found that some pattern recognition receptor (PRR) ligand adjuvants can induce a strong T-cell response (*Li et al., 2022*; *Tom et al., 2019*). PRR ligand adjuvants that target PRRs on the surface of innate immune cells, such as macrophages and dendritic cells, promote cytokine secretion and upregulate co-stimulatory molecule expression (*Didierlaurent et al., 2017*). Typical PRR adjuvants include MPLA, CpG, and MDP, which induce effective T-cell responses and have been successfully used in many vaccines such as HPV, HSV, and COVID-19 vaccines (*Fan et al., 2022*; *Del Giudice et al., 2018*; *Iwicka et al., 2022*). However, their impact on antigen processing and presentation on APCs and immunodominant epitope responses is unknown.

In this study, using MPLA, CpG, and MDP adjuvants and *H. pylori* antigens, we demonstrated that immunodominant epitopes recognized by antigen-specific T-cells were altered by adjuvants. Furthermore, we showed that the adjuvants MPLA and CpG modulate the peptide repertoires present on the APCs surface. Surprisingly, instead of revealing cryptic epitopes or presenting high-stability peptides, peptides with high binding stability for MHC-II were restrained, and low-stability peptides were presented by APCs after adjuvant treatment. The low-stability peptide presented in the adjuvant groups effectively elicits a T-cell response. Thus, altering pMHC stability in APCs provides a fundamentally new mechanism for PRR adjuvants to regulate adaptive immunity. The implications of these observations are discussed.

## Results

### Immunodominant T-cell epitope hierarchy varies in mice vaccinated with different adjuvants

To determine the influence of adjuvants on the specificity of immune responses, BALB/c mice were vaccinated with antigen UreB, an effective antigen for the *H. pylori* vaccine, combined with the adjuvants CpG, MDP, and MPLA. Then, antigen-specific T-cells from immunized mice were expanded and their interferon-γ (IFN-γ) responses to 93 UreB overlapping 18mer peptides were screened using flow cytometry. T-cells from the CpG group exhibited a dominant response to 18mer peptides U313–330 and U403–426, while T-cells from the MDP group exhibited a dominant response to U409–426 and U481–498. T-cells from the MPLA group primarily recognized U313–U330 and U505–522, whereas U481–U504 was the dominant region in T-cells from the no-adjuvant group (*Figure 1*). These data indicate that immunodominant epitopes responding to antigen-specific T-cells varied in different adjuvant vaccination groups.

### Profiling MHC-II peptides in adjuvant-treated APCs

Considering that T-cell repertoires are the same in naïve mice, we speculate that dominant epitope variations in different adjuvant vaccination groups result from alterations in APCs peptide presentation. To investigate the MHC-II–peptide repertoire presented by APCs, antigens from *H. pylori* ultrasonic supernatant were used to pulse H-2$^d$ A20 cells combined with the adjuvants CpG, MDP, and MPLA. MHC-II–peptide complexes were immunoprecipitated. Bound peptides were eluted and identified by

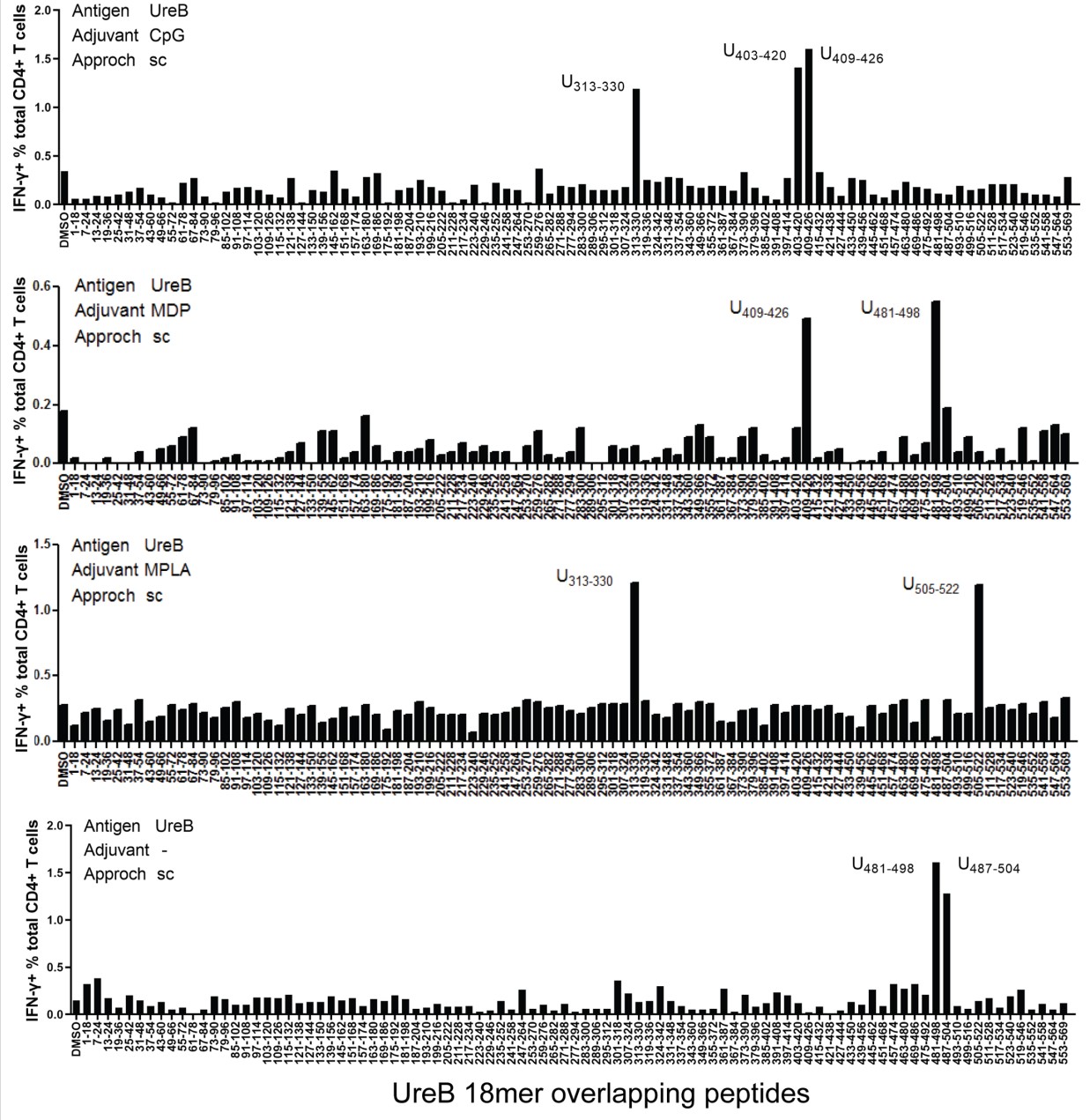

**Figure 1.** Immunodominant T-cell epitopes in different adjuvants vaccination mice. Spleens were collected from mice on day 10 post-vaccination with the antigen UreB incorporated with adjuvants CpG, MDP, and MPLA; cultured in vitro; and stimulated with a panel of overlapping UreB 18 mer peptides to assess the responsiveness of CD4$^+$ T-cells for interferon-γ (IFN-γ) using ICS. The percentages of CD4$^+$ T-cells secreting IFN-γ against each peptide were determined using flow cytometry. Locations of the dominant peptides in different groups are indicated. The results are representative of three independent experiments.

liquid chromatography–tandem MS (LC-MS/MS; *Figure 2A*). The whole proteomes of protein-pulsed A20 cells were also analyzed by LC-MS/MS to examine the effect of adjuvants on extracellular antigen phagocytosis and antigen processing- and presentation-associated protein expression.

A total of 4074 peptides were identified: 3408, 3257, 3227, and 3330 peptides in the PBS, MDP, MPLA, and CpG groups, respectively. As expected, the number and length distributions of the peptides were not influenced by the adjuvants (*Figure 2B and C*). Most identified peptides were 16 mer. Next, the binding motifs of all identified peptides were analyzed (*Figure 2D*). No obvious differences were observed following the adjuvant treatment. The binding motifs of the peptides per individual MHC-II

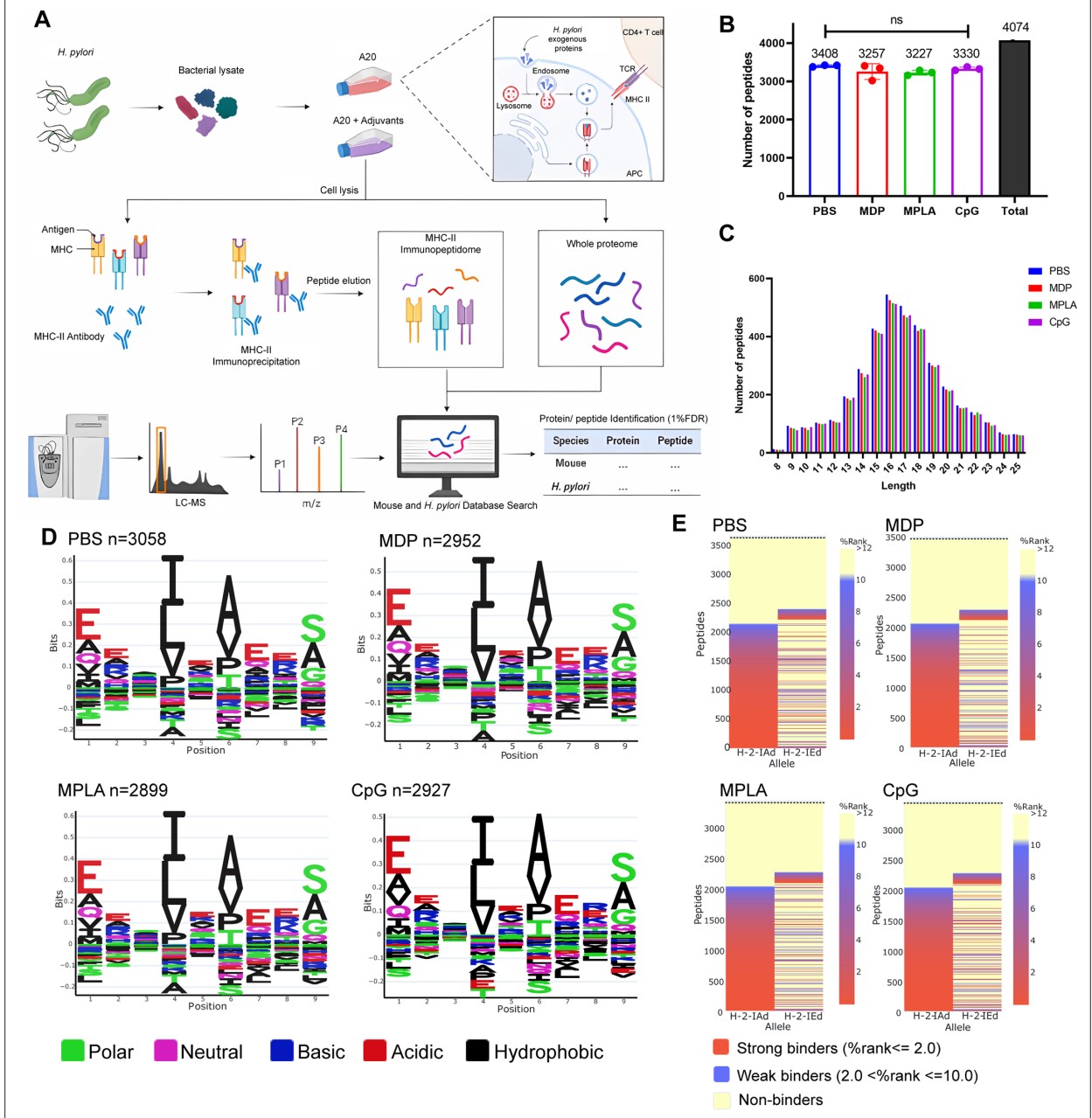

**Figure 2.** MHC-II peptidome and proteome measurements in adjuvant-treated antigen-presenting cells (APCs). A20 cells were treated with CpG ODN, MDP, or MPLA incorporated with *Helicobacter pylori* antigens for 12 hr. Most cells ($10^8$) were lysed for immunopeptidomics and the remaining cells ($10^7$) were used for proteomics. (**A**) Experimental flow chart. (**B**) Number of MHC peptides identified in the different adjuvant-treated groups (n=3, biological replicates). The numbers indicate mean values. (**C**) Length distribution of MHC peptides in different adjuvant-treated groups. (**D**) Sequence motifs of the MHC peptides identified in the adjuvant-treated groups. (**E**) Binding heatmaps of all eluted MHC peptides between 9–22 mer in adjuvant-treated groups were predicted and assigned to alleles using NetMHCIIpan. ns: not significantly different (p>0.05).

The online version of this article includes the following figure supplement(s) for figure 2:

**Figure supplement 1.** Peptide logos and MHC-II binding are assigned to individual alleles.

**Figure supplement 2.** Immunopeptidomics of J774A.1 cell line and surface marker detection.

allele, H2-IA, and H2-IE, were also analyzed (*Figure 2—figure supplement 1*). No major differences were detected. The amino acids observed at the main anchor locations were consistent with the expected binding motifs. The binding affinity data predicted using NetMHCIIpan showed that most MS-detected peptides bind to MHC-II alleles (% rank <10) and are generally assigned to the H2-IA

allele (*Figure 2E*; *Figure 2—figure supplement 1*). These results indicate that adjuvants did not affect MHC-II binding characteristics.

We also selected another H-2$^d$ cell J774A.1, a macrophage cell line, for immunopeptidome analysis in this study. Briefly, 5×10$^8$ J774A.1 cells were used for immunopeptidomics. Moreover, fewer than 350 peptides were observed at a peptide spectrum match (PSM) level of <1.0% false discovery rate (FDR). However, more than 5500 peptides were detected in 10$^8$ A20 cells at FDR <1.0% (*Figure 2—figure supplement 2A*). CD86 and MHC-II molecule expression on J774A.1 cell was substantially lower than that on A20 cells (*Figure 2—figure supplement 2B*). Low MHC-II expression on J774A.1 cell could be the reason for the lack of peptides identified by LC–MS/MS. Thus, A20 cells instead of J774A.1 cells were used for the subsequent experiments.

## Adjuvants affected the presentation of exogenous peptides exclusively

Next, we examined the exogenous peptides across the *H. pylori* genome from the MHC-II immunopeptidome. Surprisingly, *H. pylori* contained more than 3000 proteins, but less than 30 proteins were detected in the MHC-II immunopeptidomes (*Figure 3A*). To test whether the MHC-IP (MHC-immunoprecipitation) proteins had higher expression in bacteria, we ranked the individual *H. pylori* proteins according to their abundance. The abundance of MHC-IP-identified proteins in the entire *H. pylori* proteome MS data was analyzed, and we found that the abundance of these proteins was similar to that of other bacterial proteins (*Figure 3B*). We then verified whether MHC-IP proteins harbor more peptides compatible with MHC-II by comparing the ratio between the number of peptides predicted to bind MHC-II alleles and the total number of 13- to 17-mer. MHC-IP proteins contained more presentable peptides than most other bacterial proteins (*Figure 3C*).

By analyzing the host and exogenous MHC-II peptides present in different adjuvant groups, we found that 82.4% host MHC-II peptides were present in all the groups; however, only 34.4% exogenous MHC-II peptides were conserved after adjuvant treatment (*Figure 3D*; *Figure 3—figure supplement 1*). These data indicate that adjuvants affect exogenous peptide presentation. Many exogenous peptides are missing after adjuvant MPLA or CpG treatment. However, no changes were detected in the number of peptides in the host (*Figure 3E*). These data indicate that the repertoires of exogenous peptides presented by APCs are affected exclusively by adjuvants, and a smaller number of peptides are present.

## Adjuvants may affect antigen processing but not phagocytosis

To test whether the changes in exogenous peptide presentation among adjuvant groups could be explained by differences in antigen phagocytosis, we examined whole-proteome data. We observed a strong correlation between the bacterial protein abundances in different adjuvant-treated groups (*Figure 4A*). We then compared the abundance of bacterial proteins in the proteome and MHC-II immunopeptidomes and found that the bacterial protein abundance in the immunopeptidome changed significantly among the adjuvant-treated groups, and no peptides of several bacterial antigens were detected in some groups. However, only several proteins showed altered abundance in the proteome (*Figure 4B*). These results suggest that the changes in exogenous peptide presentation among the adjuvant groups cannot be explained by antigen phagocytosis.

To further investigate how adjuvants affect bacterial antigen processing and presentation, we ranked individual bacterial proteins and MHC-II peptides according to their abundance and compared them with those from the host. The overall abundance of bacterial proteins in the APCs proteome was low (*Figure 5—figure supplement 1A*). Individual bacterial protein expression was below that of most host proteins in both the PBS- and adjuvant-treated groups (*Figure 5A*). In contrast to the low abundance of bacterial proteins, the intensities of their MHC-II peptides were similar to those of peptides from the host in the PBS-treated group according to the immunopeptidome data (*Figure 5B*), indicating that bacterial peptides are preferentially present. However, in the MPLA- and CpG-treated groups, the intensities of bacterial MHC-II peptides in the immunopeptidome were much lower than those in the hosts (*Figure 5B*). These results indicate that adjuvants MPLA and CpG restrain bacterial peptide presentation.

We then investigated whole-proteome data to determine the evidence of adjuvant modification of antigen presentation. We focused on the proteins involved in antigen processing, peptidase function, ubiquitination pathway, and IFN signaling. The ubiquitination pathway and IFN signaling play crucial

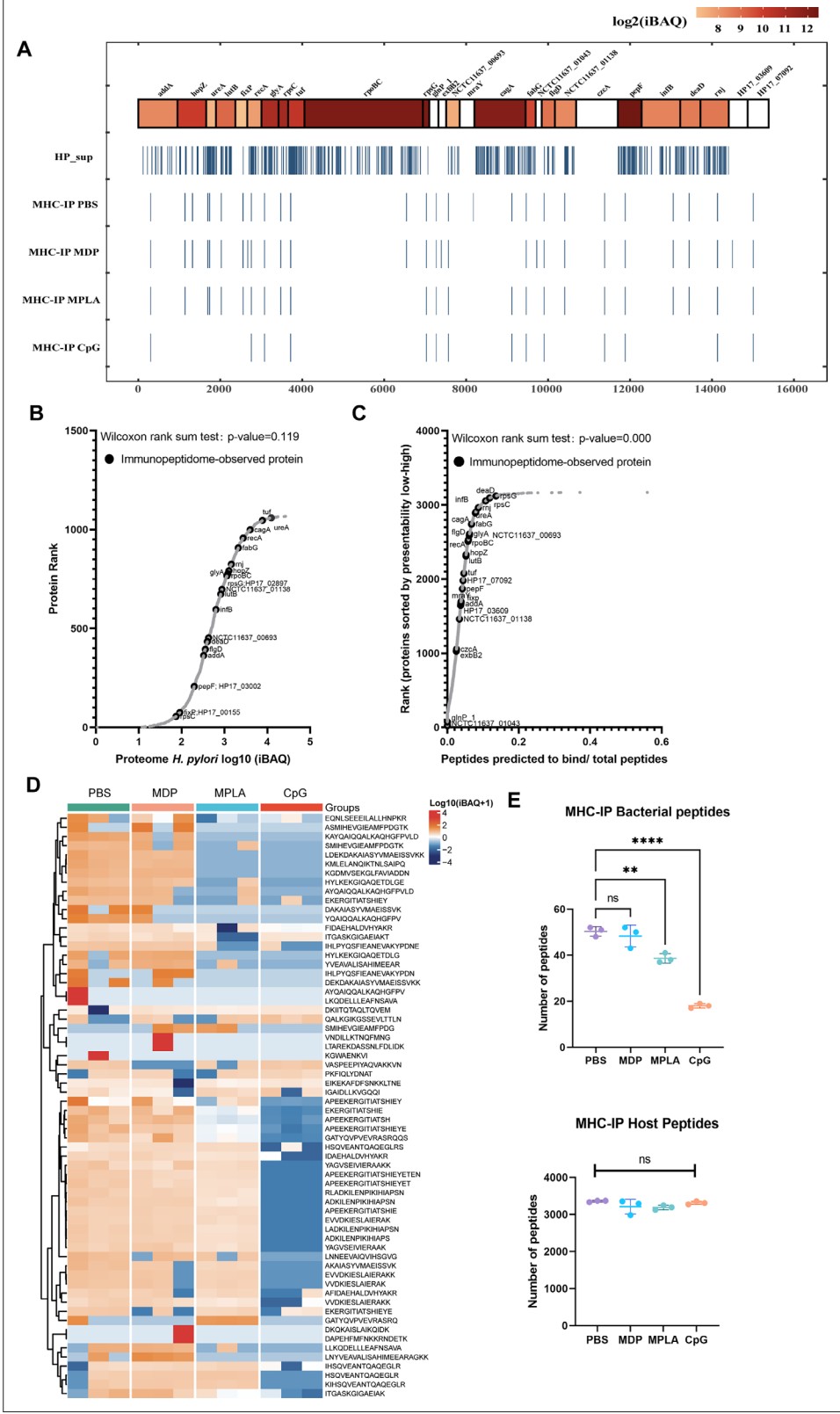

**Figure 3.** Profiling exogenous MHC-II peptides in adjuvant-treated antigen-presenting cells (APCs). (**A**) Peptide locations across the *Helicobacter pylori* genome from MHC-II immunopeptidomes. (**B**) Rank plot of each protein abundance detected in the whole-proteome of bacterial ultrasonic supernatant antigens. Proteins identified in immunopeptidomes are annotated with their respective gene names. (**C**) MHC-II presentation potential of

*Figure 3 continued on next page*

*Figure 3 continued*

bacterial proteins. All reported *H. pylori* proteins were ranked according to the ratio between the number of peptides predicted to be presented by MHC-II alleles (rank ≤2) and the total number of 13- to 17-mer. Proteins identified in immunopeptidomes are annotated with their respective gene names. (**D**) Heatmap of exogenous MHC peptides from different adjuvant groups. The identified sequences are shown. (**E**) Numbers of MHC peptides derived from bacteria and hosts were compared among different adjuvant groups. n=3. **p<0.01, ****p<0.0001.

The online version of this article includes the following figure supplement(s) for figure 3:

**Figure supplement 1.** Profiling peptides from MHC-II immunopeptidomes.

---

roles in the modification of antigen presentation and immune responses. Through KEGG enrichment analysis, we found that many proteins involved in antigen processing, peptidase function, ubiquitination pathways, and IFN signaling were altered after adjuvant treatment, particularly in the MPLA- and CpG-treated groups (*Figure 5C*; *Figure 5—figure supplement 1B*). The expression of each protein is shown in *Figure 5—figure supplement 1C* and *Supplementary file 1*. These data suggest that MPLA and CpG adjuvants may affect the antigen processing of APCs, resulting in fewer peptide presentations.

## High-stability epitopes were deficient in MPLA- and CpG-treated groups

We confirmed that the number of peptides from the exogenous antigens was significantly reduced in the MPLA- and CpG-treated groups. To further characterize the adjuvant effects on peptide presentation, the MHC-binding stability of the peptides present in the adjuvant-treated groups and that of the peptide-deficient post-adjuvant stimulation were analyzed using the IEDB website. Compared to that in the PBS-treated group, the $IC_{50}$ of the peptides binding to H2-IA and H2-IE alleles in the CpG- or MPLA-treated groups were much higher than those of the deficient peptides in the corresponding groups, which indicated that the peptides presented in the MPLA- and CpG-treated groups have lower binding stability for MHC-II (*Figure 6A*). Similar results were obtained at cutoffs of the predicted percentile rank <2 (Strong Binders) or <10 (Weak Binders) (*Figure 6—figure supplement 1*). Furthermore, we found that the peptides present in both adjuvant- and PBS-treated control groups were mainly derived from proteins tuf, recA, etc., and the deficient peptides in the MPLA- or CpG-treated group were mainly derived from proteins such as ureA and hopZ (*Figure 6B*). To validate the amino acid sequences and binding stability of MS-detected peptides, 10 peptides derived from the top four presented and deficient proteins shown in *Figure 6B*, were synthesized (*Figure 6C*). The tandem mass spectra of the synthetic peptides and experimental spectra were compared, and strong correlations between fragment ions and retention times were observed (*Figure 6D*). We then performed an MHC-II competition-binding assay to detect the binding stability of the 10 synthetic peptides in the presence of one competing MHC-II ligand. We confirmed that the peptides missing in the MPLA-/CpG-treated groups had better binding stability for MHC-II molecules than the peptides presented in the adjuvant-treated groups (*Figure 6E*). These data indicate that epitopes with high binding stability were deficient after MPLA and CpG treatment.

## Low-stability peptides presented in adjuvant-treated groups effectively induce robust T-cell responses

To evaluate whether the low-stability peptides presented in the adjuvant-treated groups could induce T-cell responses, mice were immunized with a pool of 10 synthetic peptides. Cell responses to individual peptides were detected using IFN-γ ELISpot assay on days 10 (effective phase) and 28 (memory phase) post-immunization (*Figure 7A*). We found positive responses to the low-stability peptides recA #23 (AFIDAEHALDVHYAKR) and NCTC11637-00693 #38 (IHSQVEANTQAQEGLR) as well as highly stable peptides ureA #2 (ASMIHEVGIEAMFPDGTK), ureA #3(YVEAVALISAHIMEEAR), and hopZ #53(KMLELANQIKTNLSAIPQ) on day 10 (*Figure 7B*). On day 28, the response to the low-stability peptide recA #23 (AFIDAEHALDVHYAKR) was not weaker than that of the other peptides (*Figure 7C*). Peptides with low MHC-II stability skew T-cell repertoires toward high-affinity clonotypes that have excellent responses against pathogen infection (*Baumgartner et al., 2010*; *Busch and Pamer, 1999*). To assess the functional avidities of T-cell responses induced by the low-stability

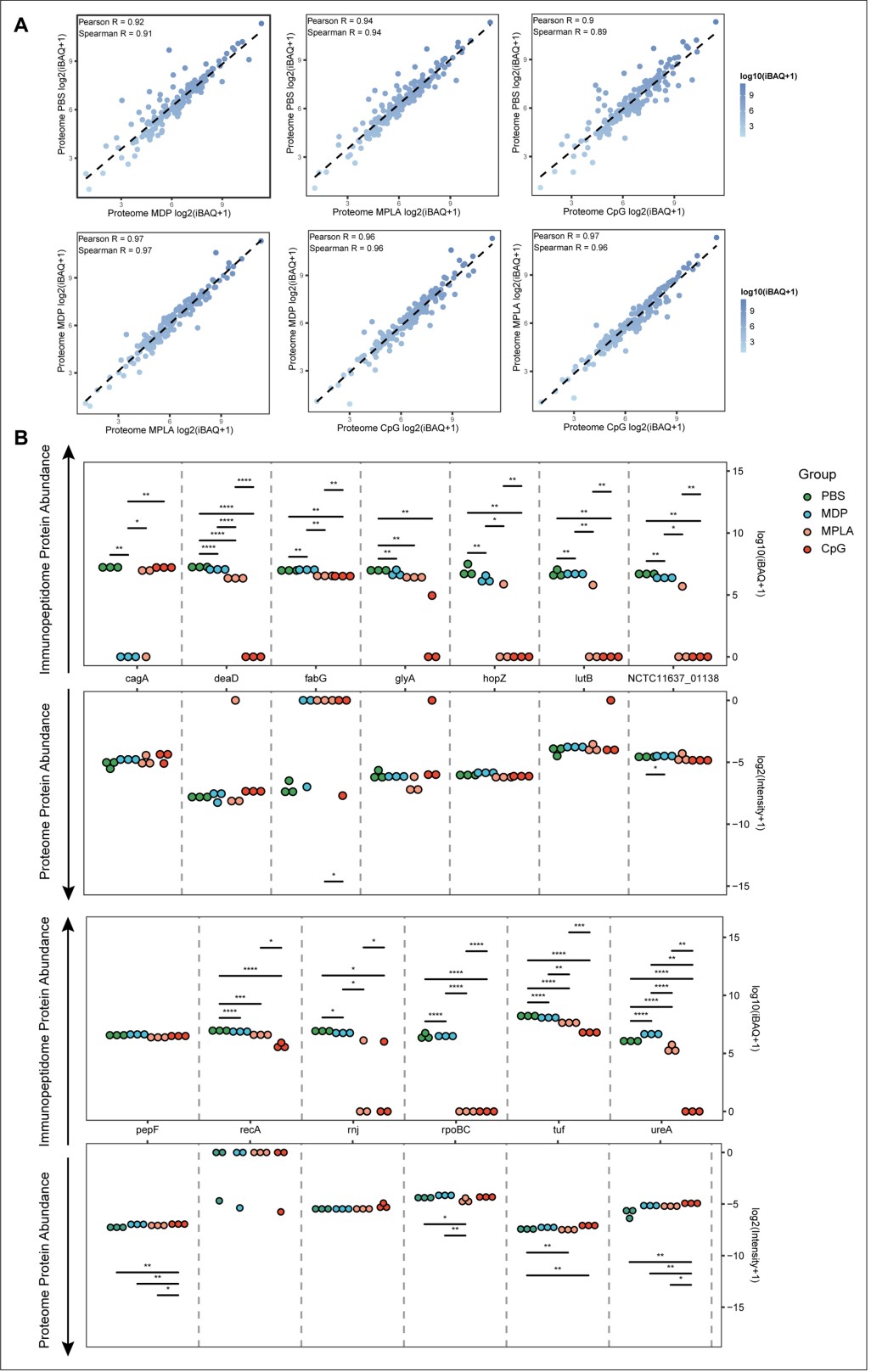

**Figure 4.** Antigen phagocytosis of antigen-presenting cells (APCs) treated with different adjuvants. (**A**) Comparison of bacterial protein abundance in APCs 12 hr post-adjuvant stimulation from whole proteomes. (**B**) Abundances of bacterial proteins from immunopeptidome and proteome were compared among adjuvant groups. *p<0.05, **p<0.01, ***p<0.001, ****p<0.0001.

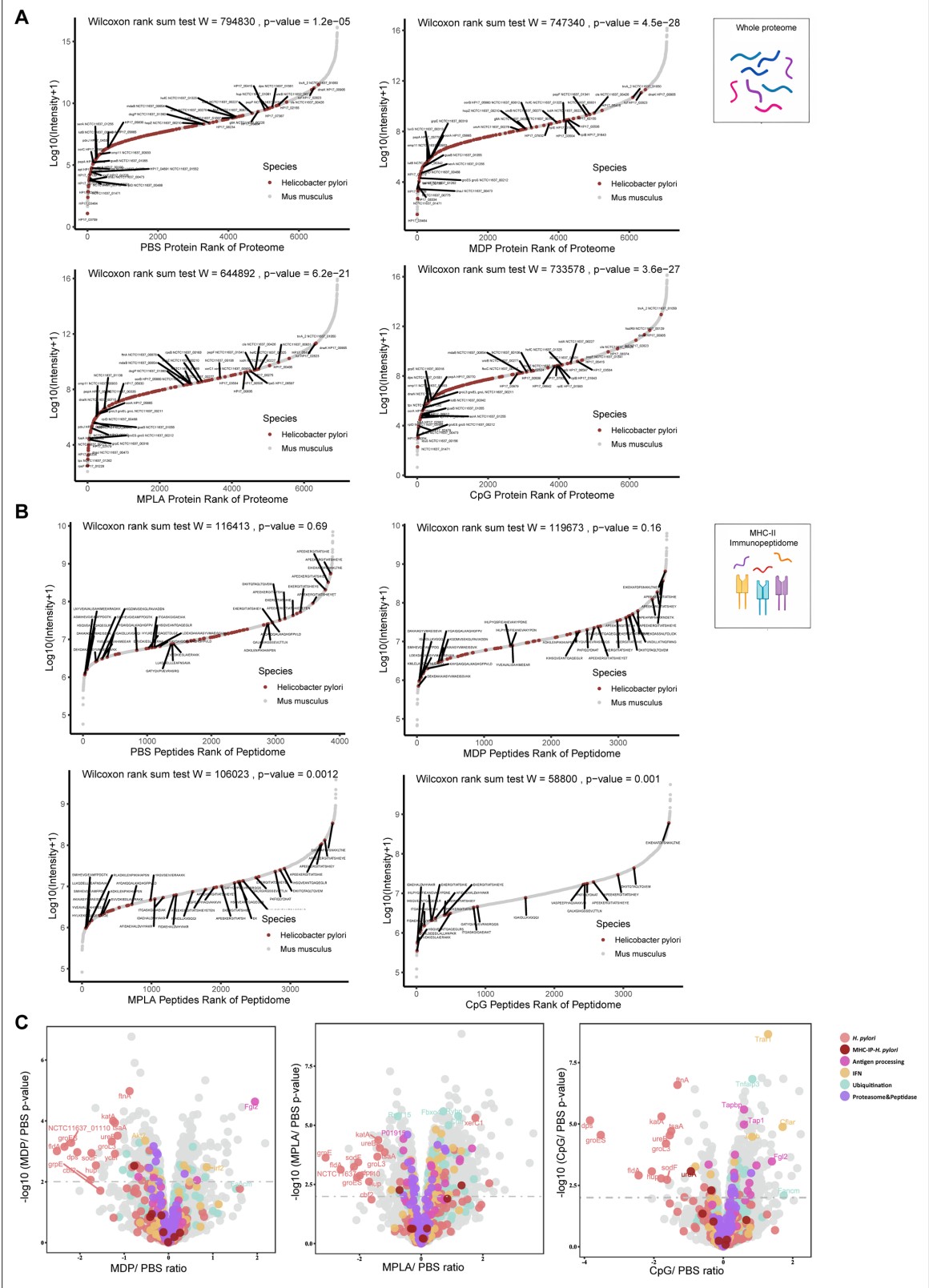

**Figure 5.** The effects of adjuvants on antigen presentation. (**A**) Rank plot of host and bacterial protein abundances in the whole-proteome and (**B**) MHC–peptide abundances from immunopeptidomes of different adjuvant groups. Bacterial proteins are marked with red, and some of them are annotated with their respective gene names. Bacterial MHC peptides are annotated with their respective amino acid sequence. (**C**) Volcano plots

*Figure 5 continued on next page*

*Figure 5 continued*

comparing protein levels between PBS- and adjuvant-treated groups in the whole-proteome. Proteins involved in antigen processing, ubiquitination, proteasome, and peptidase, and interferon (IFN) pathways are colored accordingly. Above the dashed line (p<0.01) means significant.

The online version of this article includes the following figure supplement(s) for figure 5:

**Figure supplement 1.** Bacterial proteins and host antigen presentation proteins in the whole-proteome of different adjuvant groups.

peptide presented in adjuvant-treated groups, peptide-specific CD4[+] T-cells were expanded in vitro. Moreover, the magnitude of IFN-γ responses was detected by ICS using flow cytometry on stimulation with a set of titrated peptides. We found that the low-stability peptide recA #23 induced more robust CD4[+] T-cell responses at lower peptide concentrations (*Figure 7D*). These data suggest that the low-stability peptide presented in the adjuvant-treated groups could induce robust CD4[+] T-cell responses effectively.

## Discussion

We demonstrated that adjuvants alter the specificity of the dominant Th1 epitope responses post-vaccination. Using an LC-MS/MS-based MHC-II immunopeptidome, the APCs-presented peptides of *H. pylori* antigens under the stimulation of different adjuvants were analyzed. The peptide motifs binding to MHC-II were generally consistent following stimulation with adjuvants. However, the exogenous peptide repertoires of APCs changed. Low- and high-stability peptides were presented by APCs in the control group; however, fewer peptides were detected in the adjuvant groups, and peptides with high binding stability for MHC-II presented in the control group were missing after adjuvant stimulation, particularly in the CpG group. The low-stability peptide presented in the adjuvant groups showed good immunoreactivity, effectively inducing effector and memory immune responses. Thus, our data suggested that adjuvants can restain pMHC-II stability in APCs to regulate dominant epitope-specific T-cell responses.

The protective effects of vaccines depend on the immune responses induced post-vaccination. For prophylactic vaccines, adaptive immune responses are important and restricted to a few epitopes. However, not all epitope-specific responses were protective as some epitope-induced immune responses exacerbate inflammation and autoimmune diseases (*Li et al., 2017*; *Vanderlugt and Miller, 2002*; *Mackay and Rowley, 2004*). Some epitopes induce Treg expansion and relieve inflammation (*Ooi et al., 2017*). Thus, the alteration of dominant epitope responses directly affects vaccine effectiveness. Maeda et al. characterized the modulation of adjuvants LT and LTB on antibody responses to a co-administered antigen, EDIII. They showed that LT- and LTB-adjuvanted specific antibodies displayed distinct linear epitope-binding patterns and recognized fewer peptides than the non-adjuvanted or alum-adjuvanted groups (*Maeda et al., 2017*). However, Borriello et al. showed that mannans formulated with alum as an adjuvant broadened the epitope responses of anti-spike neutralizing antibodies (*Borriello et al., 2022*). Chung et al. reported that ISCOMATRIX adjuvant promotes HA1 antibodies against large conformational epitopes spanning the receptor-binding domain, while antibodies targeting the C-terminus of the HA1 domain are generated in the unadjuvanted group. However, the number of recognized epitopes did not differ between groups (*Chung et al., 2015*). Although the capacity of adjuvants to modulate the epitope specificity of antibodies has been described, the effects of adjuvants on T-cell epitope specificity remain unclear.

Lo-Man R et al. reported that *Salmonella enterica* expressing the MalE protein induced new CD4[+] T-cell responses to peptides silent in the purified Ag with adjuvant CFA administration group (*Lo-Man et al., 2000*). Andersen et al. showed that an adenovirus vector expressing Ag85B/ESAT-6 fusion proteins induced a CD8[+] T-cell response against the ESAT-6 epitope, while a CD4[+] T-cell response to an epitope located in Ag85B was detected in the liposomal adjuvant group (*Bennekov et al., 2006*) These studies suggest that specific T-cell responses are variable and antigen delivery formulations can affect CD4[+] T-cell response specificity. In this study, using a series of synthesized overlapping peptides, we demonstrated that adjuvants modulate the hierarchy of epitope-specific Th1 responses post-vaccination.

Successful epitope-specific T-cell priming requires T-cell receptor (TCR) recognition by the pMHC on the APC surfaces. Adjuvants activate APCs and enhance co-stimulatory signals (*Yang et al., 2005*; *Hodge et al., 2005*). However, whether adjuvants can regulate the pMHC presentation on the APC

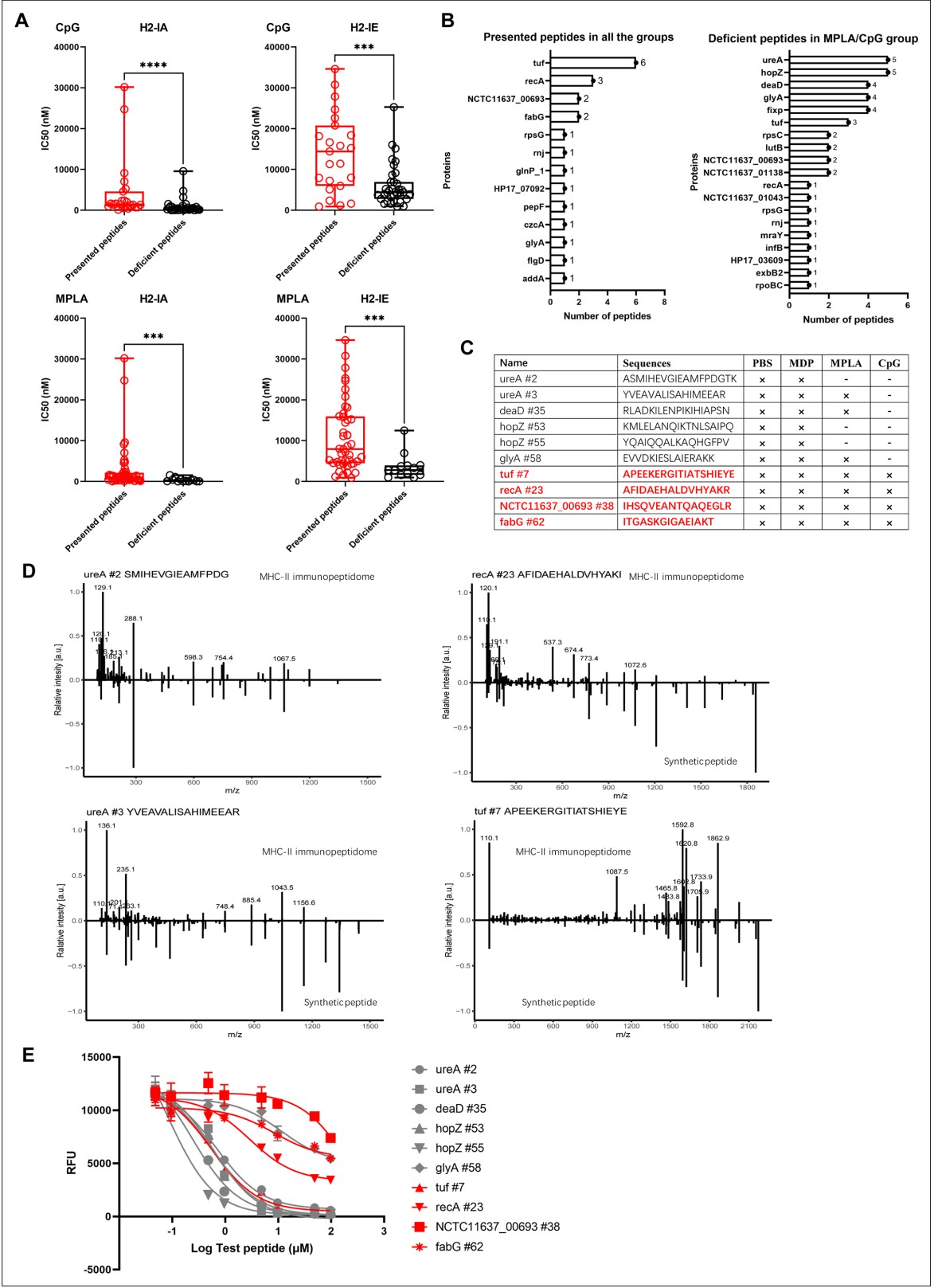

**Figure 6.** Binding affinity of MS-detected peptides was determined. (**A**) IC$_{50}$ of the presented and deficient peptides post-adjuvant stimulation from immunopeptidome binding to H2-IA and H2-IE were predicted by the NN align method using the IEDB website. The data of peptides from adjuvants MPLA and CpG are shown. High IC$_{50}$ means low binding stability. Boxes show quartiles, bars indicate medians, and whiskers show distributions. (**B**) Distribution of proteins corresponding to bacterial MHC peptides from immunopeptidome. The numbers of peptides identified by MS for each protein

*Figure 6 continued on next page*

Figure 6 continued

are indicated. (**C**) Information of 10 synthetic peptides from Top4 presented and deficient proteins. ×: Presence of peptides in the corresponding group. -: Peptides missing in the corresponding group. (**D**) Mirror plots with fragment ion mass spectra to confirm the sequences of MHC peptides from immunopeptidome. Positive y-axis, MHC-II IP sequences; negative y-axis, synthetic peptides. (**E**) Competitive binding curve of synthetic peptides for MHC-II H2-IA allele. n=3. The binding curves of peptides presented in adjuvant groups are marked with red.

The online version of this article includes the following figure supplement(s) for figure 6:

**Figure supplement 1.** Binding affinities of bacterial MHC peptides were assessed at cutoffs of percentile rank <2 (Strong Binders) or <10 (Weak Binders).

surface lacks direct evidence. In this study, using LC-MS/MS-based immunopeptidomes, we confirmed that adjuvants affected the peptide repertoires presented by MHC-II molecules on APCs, and fewer peptides were presented by APCs after MPLA and CpG stimulation. A narrow peptide repertoire increases T-cell priming possibility and improves T-cell response specificity (*Santambrogio, 2022*). Thus, we speculate that the adjuvants MPLA and CpG may enhance vaccine-induced immune responses by restricting the diverse epitope presentation of APCs rather than inducing cryptic epitope presentation. Furthermore, we found that peptides with high binding stability for MHC-II were restrained, and low-stability peptides were present on APCs after MPLA and CpG adjuvant treatment. Peptides with low MHC-II stability skew antigen-specific T-cell repertoires toward high TCR affinity clonotypes (*Baumgartner et al., 2010*). Our data explain the findings of Baumgartner et al. that MPLA and CpG adjuvants induce high TCR affinity antigen-specific T-cell clonotype responses (*Malherbe et al., 2008*). T-cell clonotypes with high TCR affinity have excellent abilities against pathogenic infections and tumors (*Busch and Pamer, 1999*; *Spear et al., 2019*; *Lima et al., 2020*). In this study, we also observed robust T-cell responses induced by the peptide with low stability through the titration assay (*Figure 7D*), although no single T-cell clonotype was used. Unfortunately, only one peptide, recA #23, with low binding stability and induced significant Th1 responses, was identified in this study. To further confirm that low-stability peptides can induce stronger and higher TCR-affinity antigen-specific T-cell clonotype responses than high-stability peptides, further studies should monitor more peptides with different stabilities.

In this study, we found that the peptide repertoires presented by APCs were significantly affected by the adjuvants CpG and MPLA, but not MDP. All three adjuvants belong to the PRR ligand adjuvant family. CpG and MPLA bind to TLRs and MDP is recognized by NOD2. Although the receptors are different, many common molecules are involved both in TLR and NLD pathway activation. Unfortunately, we did not demonstrate why the MDP had different impacts on peptide presentation compared with other adjuvants. Further investigation is required to clarify the mechanism by which MPLA, CpG, and MDP adjuvants modulate the presentation of peptides with different stabilities.

APC subtypes differ in the expression of endolysosomal proteases and DO/DM molecules (*Burster et al., 2005*; *Stoeckle et al., 2009*; *Chen et al., 2006*) and exhibit different abilities to process Ag and present peptides (*Vidard et al., 1992*; *van den Hoorn and Neefjes, 2008*). Kanellopoulos et al. reported that DCs focused HEL-specific CD4[+] T-cell responses against an I-E[d]-restricted peptide HEL103-117, while B-cells presented an additional I-A[d]-restricted peptide HEL7-31 (*Gapin et al., 1998*). Schnurr et al. showed that CD1c[+] blood DCs and MoDCs presented peptides of the antigen NY-ESO-1 on both MHC-I and MHC-II, while pDCs were limited to MHC-II presentation (*Schnurr et al., 2005*). In this study, the antigen UreB adjuvanted with different adjuvants induced antigen-specific CD4[+] T-cell responses in vivo post-vaccination, whereas no UreB peptides presented by A20 cells were detected in vitro using LC-MS/MS, possibly due to the differences in APC types involved, with many APC subtypes involved in vivo, particularly DCs, but only the B-cell line (A20) used in vitro. Another reason may be that the UreB peptide abundance was below the detection limit of LC-MS/MS in vitro.

A high abundance of pathogenic proteins is often preferred as candidate antigens in vaccine design. However, the amount of pathogen antigens were not consistent with that of antigens phagocytosed and presented by APCs. Weingarten-Gabbay et al. found that nucleocapsid protein (N) was the most abundant viral protein in SARS-CoV-2 infected A549 and HEK293T cells, but only one HLA-I peptide from N was present (*Weingarten-Gabbay et al., 2021*). Thus, antigens with high expression may be less presented than expected and induce poor T-cell responses. We found that proteins harboring more peptides compatible with the MHC-binding motifs were more likely to be present (*Figure 3C*).

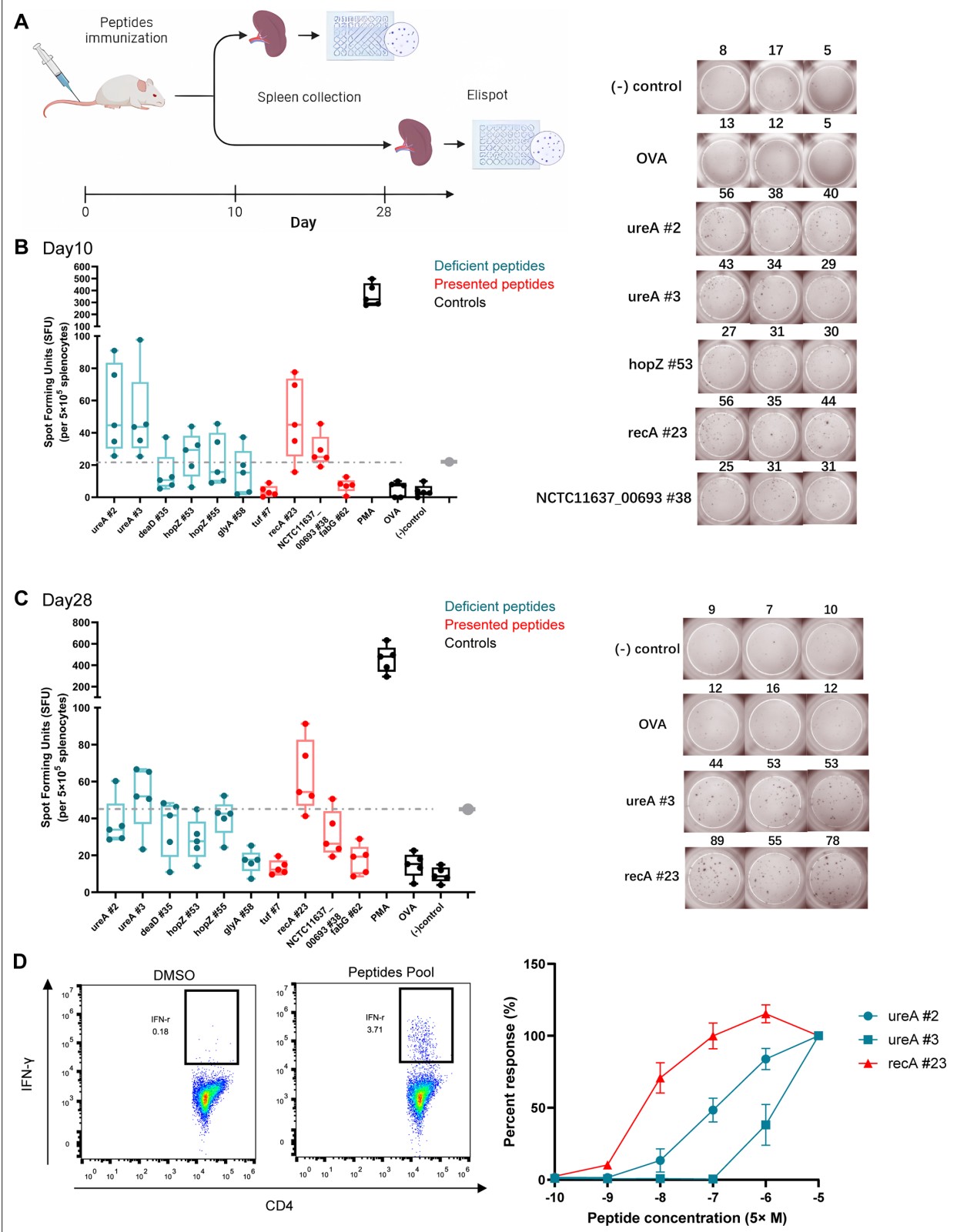

**Figure 7.** T-cell responses induced by MS-detected peptides with different binding stability were analyzed. Five BALB/c mice were immunized with a pool of 10 synthetic peptides. On days 10 and 28 post-immunization, the splenocytes were isolated and stimulated with individual peptides for interferon-γ (IFN-γ) Elisopt assay. (**A**) Experimental flow chart. (**B, C**) Elispot results on days 10 and 28. OVA peptide and non-stimulated wells were used as negative controls. PMA stimulation was used as the positive control. Responses to the peptides in the adjuvant groups are marked in red. The dashed

*Figure 7 continued*

line represents the 3x median of the OVA peptide used as the threshold for a positive response. Boxes show quartiles, bars indicate medians, and whiskers show distributions. Elispot images of positive responses from one immunized mouse are shown. Numbers indicate spot counts. (**D**) Epitope-specific CD4+ T-cells from immunized mice spleens were expanded in vitro and the IFN-γ-producing CD4+ T-cells were assessed using the peptides pool (left). The present peptide recA #23 and deficient peptides ureA #2 and ureA #3 after adjuvant treatment were titrated to restimulate the expanded cells (right, n=3). The IFN-γ responses of CD4+ T-cells were detected by FACS. The responses induced by the indicated peptides at 50 μM were considered 100%. All other responses were evaluated based on their relative strengths.

Screening dominant epitopes from all pathogen antigens is important for vaccine design. This is an advantage of immunopeptidomes based on LC-MS/MS. Several epitopes with good immunogenicity detected in this study may be used as candidates for epitope vaccine development.

Taken together, our results reveal that adjuvants influence the pMHC stability on APCs and change the epitope response specificity. In the process of adjuvant selection for vaccines, the adjuvant effect on immune response strength as well as response specificity, particularly for PRR ligand adjuvants, which mainly induce Th1- or CTL-biased responses, should be considered.

## Limitations of the study

First, the peptides screened in vitro using cell lines did not reflect the in vivo situation. The epitope presentation of APCs can be affected by the APC subtypes and inflammatory cytokines at the inoculation site (*Fiebiger et al., 2001*), which are difficult to simulate and reproduce in vitro. Second, peptide abundance below the limit of detection may be missed when using LC-MS/MS-based assays, which may cause false negatives. The possibility that these peptides induce immune responses cannot be ruled out. Third, human MHC molecules are more complex than mouse MHC molecules. Thus, whether the conclusions of this study can be generalized to the human population requires further investigation.

# Materials and methods

Key resources table

| Reagent type (species) or resource | Designation | Source or reference | Identifiers | Additional information |
|---|---|---|---|---|
| Strain, strain background (*Helicobacter pylori*) | NCTC 11637; ATCC 43504 | ATCC | Cat#: 43504 | |
| Cell line (*Mus musculus*) | A20 (B cell lymphoma line, BALB/cAnN mouse) | ATCC | Cat#: TIB-208 | |
| Cell line (*Mus musculus*) | J774A.1 (monocyte/ macrophage cell line, BALB/cN mouse) | ATCC | Cat#: TIB-67 | |
| Biological sample (*BALB/c Mouse, female*) | Primary splenic cells | VITALSTAR, China | | Freshly isolated from mouse |
| Antibody | Anti-mouse CD45-FITC (Rat monoclonal) | Biolegend | Cat#: 103108 | FACS (0.5 ul per test) |
| Antibody | Anti-mouse CD3-PE-Cy7 (Armenian Hamster monoclonal) | Biolegend | Cat#: 100320 | FACS (2 ul per test) |
| Antibody | Anti-mouse IFN-γ- PE/Dazzle594 (Rat monoclonal) | Biolegend | Cat#: 505846 | FACS (0.3 ul per test) |
| Antibody | Anti-mouse MHC-II-APC (Rat monoclonal) | Biolegend | Cat#: 107614 | FACS (1 ul per test) |
| Antibody | Anti-mouse CD4-APC (Rat monoclonal) | Biolegend | Cat#: 100412 | FACS (1 ul per test) |

*Continued on next page*

*Continued*

| Reagent type (species) or resource | Designation | Source or reference | Identifiers | Additional information |
|---|---|---|---|---|
| Antibody | Anti-mouse CD86-PE (Rat monoclonal) | Biolegend | Cat#: 105007 | FACS (5 ul per test) |
| Antibody | Anti-mouse CD80- FITC (Armenian Hamster monoclonal) | Biolegend | Cat#: 104706 | FACS (2 ul per test) |
| Antibody | Anti-mouse CD45- APC-Cy7 (Rat monoclonal) | Biolegend | Cat#: 147718 | FACS (1 ul per test) |
| Antibody | Anti-mouse CD16/32 TruStain FcX PLUS (Rat monoclonal) | Biolegend | Cat#: 156604 | FACS (1 ul per test) |
| Antibody | Anti-Mouse H2-IAd/IEd antibody (Rat monoclonal) | BioXcell | Cat#: BE00108 | IP (2 mg per sample) |
| Antibody | Anti- Mouse MHC-II (I-A/I-E) antibody (Rat monoclonal) | Thermo Fisher Scientific | Cat#: 14-5321-82 | MHC binding assay (10 µg/ ml) |
| Peptide, recombinant protein | Recombinant Murine IL-2 | PeproTech | Cat#: 212–12 | |
| Peptide, recombinant protein | Urease B subunit | This paper | Recombinant protein | Purified (purity >95%) by ourselves. |
| Peptide, recombinant protein | All peptides used in this paper | China Peptides | peptide | Customized (sequences were shown in the paper) |
| Peptide, recombinant protein | Purified mouse MHC-II | Proimmune | | Customized |
| Commercial assay or kit | Mouse IFN-γ precoated ELISpot kit | Dakewe | Cat#: 2210003 | |
| Commercial assay or kit | Cyto-Fast Fix/Perm Buffer Set | Biolegend | Cat#: 426803 | |
| Chemical compound, drug | MPLA-SM | Invivogen | Cat#: tlrl-mpla2 | |
| Chemical compound, drug | MDP | Invivogen | Cat#: tlrl-mdp | |
| Chemical compound, drug | CPG ODN | Invivogen | Cat#: tlrl-1826 | |
| Chemical compound, drug | Brefeldin A Solution | Biolegend | Cat#: 420601 | |
| Chemical compound, drug | CNBr-activated Sepharose | Cytivia | Cat#: 17-0430-01 | |
| Chemical compound, drug | CHAPS | Millipore | Cat#: 1116620001 | |
| Chemical compound, drug | Europium-Streptavidin | Abcam | Cat#: ab270228 | |
| Chemical compound, drug | Complete Freund's Adjuvant, CFA | Sigma-Aldrich | Cat#: F5881-10ML | |
| Software, algorithm | SPSS | SPSS | RRID:SCR_002865 | |
| Software, algorithm | GraphPad Prism | GraphPad Prism | RRID:SCR_002798 | |
| Software, algorithm | MHC-II binding prediction | IEDB | RRID:SCR_006604 | https://www.iedb.org/ |
| Software, algorithm | Prediction of peptide-MHC-II binding motifs | MhcVizPipe (*CaronLab, 2021*) | | https://github.com/CaronLab/MhcVizPipe/ |
| Other | Aseptic rabbit blood | Sbjbio | Cat#: SBJ-ST-RAB002 | For making *H. pylori* culture plates |

## Synthetic peptides, antibodies, and other reagents

The 18mer peptides overlapping by 12 amino acids derived from antigen UreB were synthesized and purified (purity >90%) by ChinaPeptides (Shanghai, China). All peptides were dissolved in dimethyl sulfoxide (DMSO; Sigma-Aldrich, MO, USA) and stored at −80 °C.

## Cell culture

Mouse A20 (H-2$^d$, B-cell lymphoma cell line) and J774A.1 (H-2$^d$, monocyte/macrophage cell line) cells were obtained from the American Tissue Culture Collection (ATCC, VA, USA), and confirmed to be free of mycoplasma contamination. The identity of cell lines was authenticated through STR

profiling. A20 cells were cultured in RPMI 1640 medium (Gibco) containing 10% fetal bovine serum (FBS; Biological Industries, Kibbutz Beit Haemek, Israel), 1% L-glutamine (Gibco), and 1% penicillin/ streptomycin (Gibco). J774A.1 cells were maintained in DMED (Gibco) supplemented with 10% FBS, 1% L-glutamine, and 1% penicillin/streptomycin. All cell lines were maintained at 37 °C in a humidified incubator with 5% $CO_2$.

## Mice immunization and specific T-cell bulk culture

This study was approved by the Animal Ethics Review Committee of the Eighth Affiliated Hospital of Sun Yat-Sen University.

Six- to eight-week-old SPF female BALB/c mice were immunized subcutaneously with 100 µg recombinant *H pylori* UreB (rUreB, purity >95%) emulsified in adjuvants CpG (20 µg/ mouse), MDP (30 µg/ mouse), and MPLA (10 µg/ mouse). Mice immunized with the same antigen without adjuvants were used as controls. Ten days later, the mice were euthanized with Carbon Dioxide ($CO_2$), and their spleens were harvested. Antigen-specific T-cells were expanded in vitro as described previously (*Li et al., 2015*). Briefly, the lymphocytes from spleens were isolated using a Ficoll–Hypaque (Dakewe, Shanghai, China) gradient, pulsed with 0.5 µM rUreB protein in the presence of 5 U/mL rmIL-2 (PeproTech, NJ, USA), and in vitro cultured in 'RF-10' medium consisting of RPMI 1640 (Gibco, CA, USA) supplemented with 10% fetal calf serum (Gibco), 1% L-glutamine, 1x 2-mercaptoethanol (Gibco), and 100 U/mL penicillin/ streptomycin. On day 5, the live cells were collected using a Ficoll– Hypaque gradient and cultured in a complete medium containing 20 U/mL rmIL-2. Half the medium was replaced when required.

## Flow cytometry

Bulk-cultured T-cells were stimulated with 5 µM peptides for 5 hr in the presence of brefeldin A (BioLegend). The cells were collected and stained with specific antibodies against surface markers. Intracellular cytokine staining was performed after fixation. Cells were acquired using an LSRFortessa Flow Cytometer (BD Biosciences) or a Navios Flow Cytometer (Beckman Coulter, FL, USA). The data were analyzed using FlowJo software (Tree Star, CA, USA).

## *H. pylori* lysate preparation

*Helicobacter pylori* strain NCTC 11637 (ATCC) was cultured on brain–heart infusion plates with 10% rabbit blood (Sbjbio, Nanjing, China) at 37 °C under microaerophilic conditions. Then, *H. pylori* was amplified in Brucella broth with 5% FBS (BI) at 37°C under microaerophilic conditions with gentle shaking for 24 hr. The bacteria were collected, washed, and re-suspended in PBS (Gibco) for lysis using an ultrasonic dismembrator (Biosafer, Nanjing, China). The lysates were centrifuged at 4°C and 12000×$g$ for 20 min. The supernatant was stored at −80°C for subsequent experiments.

## MHC-II complex immunoprecipitation

A20 cells were cultured in $T_{75}$ cell culture flasks (Corning, NY, USA). The expanded cells were pulsed with *H. pylori* lysates in combination with adjuvants MPLA, MDP, or CpG for 12 hr, collected, washed twice with sterile PBS, and divided into two fractions. One fraction, approximately $10^7$ cells, was used for whole-proteome analysis. The remaining $10^8$ cells were lysed in cold lysis buffer (1.0% w/v CHAPS, Protease Inhibitor tablet, and PMSF) for MHC-II complex immunoprecipitation. The cell lysates were centrifuged at 18,000×$g$ for 20 min. The supernatant (containing the MHC–peptide complexes) was transferred into a new 1.5 mL microcentrifuge tube (Corning) containing a mixture of Sepharose CNBr-activated beads (Cytivia, Utah, USA) and 2 mg Anti-Mouse H2-IA[d]/IE[d] (M5/114) antibody (BioX-cell, NH, USA). The immune complexes were captured on the beads by incubating on a rotor at 4°C for 18 hr. Sequentially, the immune complexes were transferred to a polypropylene column (Bio-Rad Laboratories, Hercules, CA, USA) and washed with 10 mL buffer A (150 mM NaCl, 20 mM Tris, pH 8.0), 10 mL buffer B (400 mM NaCl, 20 mM Tris, pH 8.0), 10 mL buffer A, and 10 mL buffer C (Tris 20 mM, pH 8.0). The MHC-II–peptide complexes were eluted with 300 µL 10% glacial acetic acid (Macklin, Shanghai, China) three times. The eluate was stored at −80°C until mass spectrometry analysis was performed.

## MHC-II peptidome LC-MS/MS data generation

MHC peptides were eluted and desalted from the beads as described previously (*Sirois et al., 2021*). The lyophilized peptides were re-suspended in ddH$_2$O containing 0.1% formic acid and 2 µL aliquots

were loaded to a nanoViper C18 (Acclaim PepMap 100, 75 µm×2 cm) trap column. Online chromatographic separation was performed using an Easy nLC 1200 system (Thermo Fisher Scientific, MA, USA). The trapping and desalting procedures were performed with 20 µL 100% solvent A (0.1% formic acid). Then, an elution gradient of 5–38% solvent B (80% acetonitrile, 0.1% formic acid) in 60 min was used on an analytical column (Acclaim PepMap RSLC, 75 µm×25 cm C18-2 µm 100 Å). Data-dependent acquisition (DDA) mass spectrometry was used to acquire tandem MS data on a Q Exactive mass spectrometer (Thermo Fisher) fitted with a Nano Flex ion source using an ion spray voltage of 1.9 kV and an interface heater temperature of 275°C. For a full mass spectrometry survey scan, the target value was $3\times10^6$ and the scan ranged from 350 to 2000 m/z at a resolution of 70,000 and a maximum injection time of 100 ms. For the MS2 scan, only spectra with a charge state of 2–5 were selected for fragmentation by high-energy collision dissociation with a normalized collision energy of 28. The MS2 spectra were acquired in the ion trap in rapid mode with an AGC target of 8000 and a maximum injection time of 50 ms. The dynamic exclusion was 25 s.

## Whole-proteome LC-MS/MS data generation

Protein aliquots were mixed with 200 µL 8 M urea in Nanosep Centrifugal Devices (PALL) and centrifuged at 12,000×$g$ at 20°C for 20 min. All the centrifugation steps were performed under the same conditions, allowing for maximal concentration. Then, 200 µL 8 M urea solution with 10 mM DTT was added, and the reduction reaction was performed for 2 hr at 37°C. The solution was removed by centrifugation and 200 µL 8 M urea solution with 50 mM iodoacetamide (IAA) was added. The samples were incubated in the dark for 15 min at room temperature. The ultra-fraction tube was washed with 200 µL 8 M urea three times and 200 µL 25 mM ammonium bicarbonate three times by centrifugation at 12,000×$g$ for 20 min at room temperature. Then, 100 µL 25 mM ammonium bicarbonate containing 0.01 µg/µL trypsin was added to each filter tube and incubated at 37°C for 12 hr. The filter tubes were washed twice with 100 µL 25 mM ammonium bicarbonate by centrifugation at 12,000×$g$ for 10 min. The flow-through fractions were collected and lyophilized.

The lyophilized peptides were re-suspended in ddH$_2$O containing 0.1% formic acid, and 2 µL aliquots were loaded to a nanoViper C18 (Acclaim PepMap 100, 75 µm×2 cm) trap column. Online chromatographic separation was performed using an Easy nLC 1200 system (Thermo Fisher). The trapping and desalting procedures were carried out with 20 µL 100% solvent A (0.1% formic acid). Then, an elution gradient of 5–38% solvent B (80% acetonitrile, 0.1% formic acid) in 60 min was used on an analytical column (Acclaim PepMap RSLC, 75 µm×25 cm C18-2 µm 100 Å). A TimsTof Pro2 mass spectrometer (Bruker, USA) fitted with a Bruker captive spray ion source was operated in DIA-PASEF mode with a scan range of 100–1700 m/z and 10 PASEF ramps. The TIMS settings were 100ms ramp and accumulation time (100% duty cycle) and 9.42 Hz ramp rate; this resulted in 1.8 s cycle time and setting at a 5000 absolute intensity threshold. The collision energy remained at the default with a base of 1.60 /K0[V s/cm$^2$] set at 59 eV and a base of 0.60 /K0[Vs/cm$^2$] set at 20 eV. Active exclusion was enabled with a 0.4 min release. TIMS ranges were initially set from one range of 0.60–1.60 /K0[V s/cm$^2$], as seen in most published studies.

## Peptide identification verification

The screened peptides were verified using synthetic peptides. Peptides were synthesized by China-Peptides (Shanghai, China) at a purity >90% and dissolved to 10 mM with DMSO. For LC-MS/MS measurements, the peptides were pooled and further diluted with 0.1% FA/3% ACN to load 120 fmol/mL on the column. LC-MS/MS measurements were performed as previously described. The experimental and synthetic sequences were confirmed by plotting fragment ion mass spectra.

## MHC-II–peptide binding assay in vitro

A competition assay based on the binding of a high-affinity biotin-labeled control peptide (biotin–(Ahx)–(Ahx)–YAHAAHAAHAAHAAHAA) to MHC-II molecules was used to test peptide binding to MHC-II molecules. The assays were performed as previously described (*Salvat et al., 2014*). Briefly, test peptides were diluted to a series of concentrations and co-incubated with 0.1 µM biotin-labeled peptide and 50 nM purified MHC-II proteins (ProImmune, Oxford, UK) at 37 °C for 24 hr in the presence of octyl-β-D-glucopyranaside (Sigma-Aldrich). The binding reaction was neutralized and the products were transferred to MHC Class II (I-A/I-E) monoclonal antibody (M5/114.15.2) (Thermo Fisher)-coated

ELISA plates (Corning), which were incubated at 4 °C overnight. Finally, diluted europium–streptavidin (Abcam, MA, USA) was added to each well of the ELISA plate, and the fluorescence was read using a time-resolved fluorescent plate reader (Thermo Fisher Scientific, MA, USA) with europium settings. Each peptide was tested at eight concentrations in three independent experiments.

### Peptide immunization and ELISpot assay

Six- to eight-week-old SPF female BALB/c mice were immunized subcutaneously with a pool of synthetic peptides (50 µg for each peptide) emulsified in Complete Freunds Adjuvant (Sigma-Aldrich). At 10 and 28 days post-vaccination, the mice were euthanized, and the spleens were removed for ELISpot assays.

Splenocytes (500,000 cells/well) were treated with red blood cell lysis buffer and stimulated with 5 µM peptides in triplicate in ELISpot plates (Dakewe Biotech, Shenzhen, China) for 18 hr. IFN-γ secretion was detected using capture and detection antibodies and imaged using an ELISpot & FluoroSpot Reader (Mabtech, Stockholm, Sweden). OVA peptide$_{323–339}$ (ISQAVHAAHAEINEAGR) and non-stimulated wells were used as negative controls. PMA (BioGems, NJ, USA) was used as a positive control. A threefold increase over baseline was used as the threshold for positive responses.

### Peptide–MHC-II binding motif prediction

The peptide–MHC-II binding motifs, alignment, and peptide clustering were predicted using the MhcVizPipe (MVP) software tool (*CaronLab, 2021*, https://github.com/CaronLab/MhcVizPipe). The MHC-II alleles I-A$^d$ or I-E$^d$ were selected while maintaining all standard parameters.

### MHC-II binding prediction

The online tools NetMHCIIpan 4.1, SMM alignment, and NN_ alignment in IEDB (https://www.iedb.org/) were used to predict the binding of peptides to MHC-II alleles. To test whether MS-detected proteins harbored more peptides compatible with MHC-II binding motifs, all *H. pylori* strain NCTC 11637 protein sequences were retrieved from the UniProt database, and the ratio between the number of peptides predicted to bind MHC-II alleles and the total number of 13- to 17-mer at a cutoff of predicted percentile rank values (%) <2 were compared.

### Statistical analysis

Data are shown as mean ± SD. One-way analysis of variance (ANOVA) was performed to compare statistical significance among three or more groups. Student's *t*-test was used to compare the differences between the two groups; however, when the variances differed, the Mann–Whitney U test was used. The chi-square test was used to analyze the differences in the constituent ratios between the two groups. Pearson's correlation was used to test the correlation between two continuous variables, and Spearman's correlation was used for categorical variables. SPSS statistical software (version 25; SPSS Inc, IL, USA) and GraphPad Prism (version 9.0; GraphPad Software, CA, USA) were used for the statistical analyses. $p < 0.05$ was considered statistically significant.

## Acknowledgements

We thank Miss. Yunyun Shi for her help with the proteome and peptidome analyses. This study was supported by the National Nature Science Foundation of China No. 82001751 (BL) and the Futian Healthcare Research Project No. FTWS2021061 (BL); No. FTWS2022063 (BL); No. FTWS028 (BL).

## Additional information

### Funding

| Funder | Grant reference number | Author |
| --- | --- | --- |
| National Nature Science Foundation of China | 82001751 | Bin Li |

| Funder | Grant reference number | Author |
|---|---|---|
| Futian Healthcare Research Project | FTWS2021061 | Bin Li |
| Futian Healthcare Research Project | FTWS2022063 | Bin Li |
| Futian Healthcare Research Project | FTWS028 | Bin Li |

The funders had no role in study design, data collection and interpretation, or the decision to submit the work for publication.

### Author contributions

Bin Li, Conceptualization, Resources, Data curation, Formal analysis, Supervision, Funding acquisition, Validation, Investigation, Visualization, Methodology, Writing – original draft, Project administration, Writing – review and editing; Jin Zhang, Formal analysis, Investigation, Visualization, Methodology; Taojun He, Investigation, Methodology; Hanmei Yuan, Methodology; Hui Wu, Investigation; Peng Wang, Conceptualization, Resources, Supervision, Project administration, Writing – review and editing; Chao Wu, Conceptualization, Resources, Supervision, Validation, Writing – original draft, Project administration, Writing – review and editing

### Author ORCIDs

Bin Li https://orcid.org/0000-0002-8429-7109
Peng Wang https://orcid.org/0000-0002-5592-6842

### Ethics

This study was approved by the Animal Ethics Review Committee of the Eighth Affiliated Hospital of Sun Yat-Sen University (Permit Number: 2021-003-01).

Reviewer #1 (Public Review): https://doi.org/10.7554/eLife.99173.3.sa1
Reviewer #2 (Public Review): https://doi.org/10.7554/eLife.99173.3.sa2
Author response https://doi.org/10.7554/eLife.99173.3.sa3

## Additional files

### Supplementary files

• Supplementary file 1. Whole-proteome data, related to *Figure 5* and *Figure 5—figure supplement 1*. The whole proteome data of A20 cells treated with different adjuvants at 12 hr. The '1 a' sheet contains the KEGG and GO annotations of proteins from antigen processing, peptidase function, ubiquitination pathway, interferon (IFN) signaling, and *H. pylori*. The '1b' sheet contains the iBAQ values of each protein in different adjuvant groups.

• MDAR checklist

### Data availability

All data generated or analysed during this study are included in the manuscript and supporting files; supplementary file has been provided for Figures 5 and Figure 5—figure supplement 1. The original mass spectra and peptide spectrum matches have been deposited in iProX under accession number IPX0007611000.

The following dataset was generated:

| Author(s) | Year | Dataset title | Dataset URL | Database and Identifier |
|---|---|---|---|---|
| Li B | 2024 | Adjuvant MHC-II Immunopeptidome and proteome | https://www.iprox.cn/page/project.html?id=IPX0007611000 | iProX, IPX0007611000 |

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
