## [Editor Report · eLife Assessment]

This **important** study provides interesting insights into the mechanisms of action of adjuvants. It shows that adjuvants, MPLA and CpG especially, modulate the peptide repertoires presented on the surface of antigen presenting cells, and surprisingly, adjuvant favored the presentation of low-stability peptides rather than high-stability peptides by antigen presenting cells. As a result, the low stability peptide presented in adjuvant groups elicits T cell response effectively. Evidence in support of these conclusions is **solid**, and this paper would be of interest to vaccinologists and immunologists.

---

## [Referee Report · Reviewer #1 (Public Review)]

Summary:

Li et al investigated how adjuvants such as MPLA and CpG influence antigen presentation at the level of the Antigen presenting cell and MHCII : peptide interaction. They found that use of MPLA or CpG influences the exogenous peptide repertoire presented by MHC II molecules. Additionally, their observations included the finding that peptides with low-stability peptide:MHC interactions yielded more robust CD4+ T cell responses in mice. These phenomena were illustrated specifically for 2 pattern recognition receptor activating adjuvants. This work represents a step forward for how adjuvants program CD4+ Th responses and provide further evidence regarding expected mechanisms of PRR adjuvants in enhancing CD4+ T cell responses in the setting of vaccination.

Strengths:

The authors use a variety of systems to analyze this question. Initial observations were collected in an H pylori model of vaccination with a demonstration of immunodominance differences simply by adjuvant type, followed by analysis of MHC:peptide as well as proteomic analysis with comparison by adjuvant group. Their analysis returns to peptide immunization and analysis of strength of relative CD4+ T cell responses, through calculation of IC:50 values and strength of binding. This is a comprehensive work. The logical sequence of experiments makes sense and follows an unexpected observation through to trying to understand that process further with peptide immunization and its impact on Th responses. This work will premise further studies into the mechanisms of adjuvants on T cells

Weaknesses:

While MDP has a different manner of interaction as an adjuvant compared to CpG and MPLA, it is unclear why MDP has a different impact on peptide presentation and it should be further investigated, or at minimum highlighted in the discussion as an area that requires further investigation.

It is alluded by the authors that TLR activating adjuvants mediate selective, low affinity, exogenous peptide binding onto MHC class II molecules. However, this was not demonstrated to be related specifically to TLR binding. Wonder if some work with TLR deficient mice (TLR 4KO for example) could evaluate this phenomenon more specifically

Lastly, it is unclear if the peptide immunization experiment reveals a clear pattern related to high and low stability peptides among the peptides analyzed.

---

## [Referee Report · Reviewer #2 (Public Review)]

Adjuvants boost antigen-specific immune responses to vaccines. However, whether adjuvants modulate the epitope immunodominance and the mechanisms involved in adjuvant's effect on antigen processing and presentation are not fully characterized. In this manuscript, Li et al report that immunodominant epitopes recognized by antigen-specific T cells are altered by adjuvants.

Using MPLA, CpG, and MDP adjuvants and H. pylori antigens, the authors screened the dominant epitopes of Th1 responses in mice post-vaccination with different adjuvants and found that adjuvants altered antigen-specific CD4+ T cell immunodominant epitope hierarchy. They show that adjuvants, MPLA and CpG especially, modulate the peptide repertoires presented on the surface of APCs. Surprisingly, adjuvant favored the presentation of low-stability peptides rather than high-stability peptides by APCs. As a result, the low stability peptide presented in adjuvant groups elicits T cell response effectively.

---

## [Author Response]

**Reviewer #1 (Public Review):**
Summary:Li et al investigated how adjuvants such as MPLA and CpG influence antigen presentation at the level of the Antigen-presenting cell and MHCII : peptide interaction. They found that the use of MPLA or CpG influences the exogenous peptide repertoire presented by MHC II molecules. Additionally, their observations included the finding that peptides with low-stability peptide:MHC interactions yielded more robust CD4+ T cell responses in mice. These phenomena were illustrated specifically for 2 pattern recognition receptor activating adjuvants. This work represents a step forward for how adjuvants program CD4+ Th responses and provides further evidence regarding the expected mechanisms of PRR adjuvants in enhancing CD4+ T cell responses in the setting of vaccination.Strengths:The authors use a variety of systems to analyze this question. Initial observations were collected in an H pylori model of vaccination with a demonstration of immunodominance differences simply by adjuvant type, followed by analysis of MHC:peptide as well as proteomic analysis with comparison by adjuvant group. Their analysis returns to peptide immunization and analysis of strength of relative CD4+ T cell responses, through calculation of IC:50 values and strength of binding. This is a comprehensive work. The logical sequence of experiments makes sense and follows an unexpected observation through to trying to understand that process further with peptide immunization and its impact on Th responses. This work will premise further studies into the mechanisms of adjuvants on T cells.Weaknesses:Comment 1. While MDP has a different manner of interaction as an adjuvant compared to CpG and MPLA, it is unclear why MDP has a different impact on peptide presentation and it should be further investigated, or at minimum highlighted in the discussion as an area that requires further investigation.

Thank you for the suggestion. We investigated the reasons for the different effects of MDP on peptide presentation compared with those of CpG and MPLA. We found that the expression of some proteins involved in antigen processing and presentation, such as CTSS, H2-DM, Ifi30, and CD74, was substantially lower in the MDP-treated group than in the CpG- and MPLA-treated groups. To further confirm whether these proteins play a key role during adjuvant modification of peptide presentation, we knocked down them using shRNA and then performed immunopeptidomics. The original mass spectra and peptide spectrum matches have been deposited in the public proteomics repository iProX (https://www.iprox.cn/page/home.html) under accession number IPX0007611000. Unfortunately, the expected results for peptide presentation repertoires were not observed. Thus, we hypothesized that the different effects of MDP on peptide presentation might not result from differences in protein expression. We cannot exclude the possibility that some other proteins that may be important in this process were overlooked. We are still working on the mechanisms and do not have an exact conclusion. Thus, we did not present related data in this manuscript.

The related statements were added in the Discussion section on page 13, lines 292–299: “In this study, we found that the peptide repertoires presented by APCs were significantly affected by the adjuvants CpG and MPLA, but not MDP. All three adjuvants belong to the PRR ligand adjuvant family. CpG and MPLA bind to TLRs and MDP is recognized by NOD2. Although the receptors are different, many common molecules are involved both in TLR and NLD pathway activation. Unfortunately, we did not demonstrate why the MDP had different impacts on peptide presentation compared with other adjuvants. Further investigation is required to clarify the mechanism by which MPLA, CpG, and MDP adjuvants modulate the presentation of peptides with different stabilities.”

Comment 2. It is alluded by the authors that TLR activating adjuvants mediate selective, low affinity, exogenous peptide binding onto MHC class II molecules. However, this was not demonstrated to be related specifically to TLR binding. I wonder if some work with TLR deficient mice (TLR 4KO for example) could evaluate this phenomenon more specifically.

Thank you for the suggestion. This is an important point that was overlooked in this study. Based on published research on the mechanisms of PRR adjuvants, CpG and MPLA, we believe that the effect of CpG and MPLA on APCs-selective epitope presentation needs to be bound to the corresponding receptor, although we did not give a definitive conclusion in the manuscript.

To confirm the TLR-activating adjuvants affecting peptides presented on MHC molecules specifically through TLR binding, we have used CRISPR-cas9 to knock out TLR4 and TLR9 of A20 cells and repeated the experiments, as suggested. We chose TLR4- and TLR9- knockout A20 cell lines instead of TLR-deficient mice because a large number of APCs are required for immunopeptidomics. Moreover, the data observed in this study were based on the A20 cell line. However, these experiments are time-consuming. Unfortunately, we were unable to provide timely data. In addition, we believe that elucidating the downstream molecular mechanisms of TLR activation is necessary, as mentioned in comment 1. All these data will be combined and reported in our upcoming publications.

Comment 3. It is unclear to me if this observation is H pylori model/antigen-specific. It may have been nice to characterize the phenomenon with a different set of antigens as supplemental. Lastly, it is unclear if the peptide immunization experiment reveals a clear pattern related to high and low-stability peptides among the peptides analyzed.Q1: It is unclear to me if this observation is H. pylori model/antigen-specific. It may have been nice to characterize the phenomenon with a different set of antigens as supplemental.

Thank you for the comment. To confirm the effect of the adjuvant on the exogenous peptide repertoire presented by MHC II molecules, a set of antigens from another bacterium, *Pseudomonas aeruginosa*, was used, and the experiments were repeated. The A20 cells were treated with CpG and pulsed with *Pseudomonas aeruginosa* antigens. Twelve hours later, MHC-II–peptide complexes were immunoprecipitated, and immunopeptidomics were performed. The data are shown below (Author response image 1). Information on the MHC-peptides from *Pseudomonas aeruginosa* is given in the Supplementary Table named “Table S3 Response to comment3”. A total of 713 and 205 bacterial peptides were identified in the PBS and CpG groups (Author response image 1A). The number of exogenous peptides in the CpG-treated group was significantly lower than that in the PBS-treated control group (Author response image 1B). A total of 568 bacterial peptides were presented only in the PBS group; 60 bacterial peptides were presented in the CpG-treated group, and 145 bacterial peptides were presented in both groups (Author response image 1C). We then analyzed the MHC-binding stability of the peptides present in the adjuvant-treated group and that of the peptide-deficient after adjuvant stimulation using the IEDB website. We found that the IC50 of the peptides in the adjuvant-treated group were much higher than those of the deficient peptides, which indicated that the peptides presented in the CpG-treated groups have lower binding stability for MHC-II (Author response image 1D). These results indicate that CpG adjuvant affects the presentation of exogenous peptides with high binding stability, which is consistent with the data reported in our manuscript. Using another set of antigens, we confirmed that our observations were not H. pylori model- or antigen-specific.

**Author response image 1. sa3fig1:** MHC-II peptidome measurements in adjuvant-treated APCs pulsed with *Pseudomonas aeruginosa* antigens. (A) Total number of bacterial peptides identified in the PBS- and CpG-treated groups. (B) The number and length distribution of bacterial peptides in different groups were compared. (C) Venn diagrams showing the distribution of bacterial peptides in different groups. (D) IC50 of the presented, deficient, and co-presented peptides post-adjuvant stimulation from immunopeptidome binding to H2-IA and H2-IE were predicted using the IEDB website. High IC50 means low binding stability. *p<0.05, **p<0.01.

Q2: Lastly, it is unclear if the peptide immunization experiment reveals a clear pattern related to high and low-stability peptides among the peptides analyzed.

In this study, we used a peptide immunization experiment to evaluate the responses induced by the screened peptides with different stabilities. In addition to this method, tetramer staining and ELISA have been used to assess epitope-specific T-cell proliferation and cytokine secretion. Among these, tetramer staining is often used in studies involving model antigens. However, as many peptides were screened in our study, synthesizing a sufficient number of tetramers was difficult. However, we believe that the experimental data obtained in this study support the conclusion. Nevertheless, we agree that more methods applied will make the pattern more clearly.

**Reviewer #2 (Public Review):**
Adjuvants boost antigen-specific immune responses to vaccines. However, whether adjuvants modulate the epitope immunodominance and the mechanisms involved in adjuvant's effect on antigen processing and presentation are not fully characterized. In this manuscript, Li et al report that immunodominant epitopes recognized by antigen-specific T cells are altered by adjuvants.Using MPLA, CpG, and MDP adjuvants and H. pylori antigens, the authors screened the dominant epitopes of Th1 responses in mice post-vaccination with different adjuvants and found that adjuvants altered antigen-specific CD4+ T cell immunodominant epitope hierarchy. They show that adjuvants, MPLA and CpG especially, modulate the peptide repertoires presented on the surface of APCs. Surprisingly, adjuvant favored the presentation of low-stability peptides rather than high-stability peptides by APCs. As a result, the low stability peptide presented in adjuvant groups elicits T cell response effectively.

Thanks a lot for your comments.

**Reviewer #1 (Recommendations For The Authors):**
Recommendation 1. Figure 6: The peptides considered low affinity- it would be helpful to specify from which adjuvant they were collected from. When they are pooled it is unclear if we are analyzing peptides collected from adjuvanting with any of the three adjuvants studied.

Thank you for the suggestion. The related description in Figure 6 has been modified in the revised manuscript. Data for the peptides identified from the adjuvants MPLA- and CpG-treated groups are shown separately.

Recommendation 2. It is unclear to me why the A20 cell line is less preferred to the J774 line for the immunopeptidome analysis - can the authors expand on this?

We apologize for not clearly explaining this in the original manuscript. In fact, the A20 cell line is better than J774A.1 cell line for immunopeptidomics experiments. Compared to J774A.1 cells, more MHC-II peptides were obtained from a smaller number of A20 cells using immunopeptidomics. At the beginning of this study, we chose the J774A.1 cell line as it is a macrophage cell line. J774A.1 cells (up to 5×108) were pulsed with the antigens, and MHC-II–peptide complexes were eluted from the cell surface for immunopeptidomics. Unfortunately, only a few hundred peptides from the host were detected and no exogenous peptides were detected. Next, we tested the A20 cell line. In total, 108 A20 cells were used in this study. More than 3500 host peptides and approximately 50 exogenous peptides have been identified. These data indicate that the A20 cell line was better.

To investigate the reasons for this, we detected MHC-II expression on cell surfaces using FACS. Our purpose was to elute peptides from MHC–peptide complexes present on the cell surface. Low MHC expression resulted in the elution of a few peptides. We found the MFI of MHC-II molecules on J774A.1 cell is about 500; however, the MFI of MHC-II molecules on A20 cells is more than 300,000. These data indicate that MHC-II expression on A20 cells was much higher than that on J774A.1 cells. J774A.1 cell is a macrophage cell line. Macrophages have excellent antigen phagocytic capabilities; however, their ability to present antigens is relatively weak. MHC molecules on the macrophage cell surface can be upregulated in the stimulation of some cytokines, for example, IFN-γ. In this study, we used adjuvants as stimulators and did not want to use additional cytokine stimulators. Thus, J774A.1 cells were not used in the present study.

The related statements are reflected on page 6 lines 120–128 “We also selected another H-2d cell J774A.1, a macrophage cell line, for immunopeptidome analysis in this study. Briefly, 5×108 J774A.1 cells were used for immunopeptidomics. Moreover, fewer than 350 peptides were observed at a peptide spectrum match (PSM) level of < 1.0% false discovery rate (FDR). However, more than 5500 peptides were detected in 108 A20 cells at FDR < 1.0% (Figure S2A). CD86 and MHC-II molecule expression on J774A.1 cells was substantially lower than that on A20 cells (Figure S2B). Low MHC-II expression on J774A.1 cells could be the reason for the lack of peptides identified by LC–MS/MS. Thus, A20 cells instead of J774A.1 cells were used for the subsequent experiments.”

Recommendation 3. Lines 172-177, can more details be provided about the whole proteome analysis? The plots are shown for relative representation of protein expression to PBS, but it is unclear to me what examples of these proteins are (IFN pathway, Ubiquitination pathway). Could these be confirmed by protein expression analyses in supplemental?

Thank you for the suggestion. In this study, we conducted whole proteome analysis to investigate changes in protein expression across different pathways in the adjuvant groups. Through KEGG enrichment analysis, we compared the differential expression of MHC presentation pathway proteins (such as H2-M, Ifi30, CD74, CTSS, proteasome, and peptidase subunits) between the PBS- and adjuvant-treated groups using our proteome data. In addition, we focused on IFN and ubiquitination pathways that play crucial roles in antigen presentation modification and immune response. The proteins and their relative expression in these pathways are shown in Figure S4B. Details regarding the protein names and expressions are provided in Supplemental Table S2 of the revised manuscript.

The original statements in the results “Then, we analyzed the whole proteome data to determine whether the proteins involved in antigen presentation and processing were altered. We found that proteins involved in antigen processing, peptidase function, ubiquitination pathway, and interferon (IFN) signaling were altered post adjuvants treatment, especially in MPLA and CpG groups (Figure 5C; Figure S4B and S4C). These data suggest that adjuvants MPLA and CpG may affect the antigen processing of APCs, resulting in fewer peptides presentation.” This has been revised on page 8 lines 172–182 as “We then investigated whole-proteome data to determine the evidence of adjuvant modification of antigen presentation. We focused on the proteins involved in antigen processing, peptidase function, ubiquitination pathway, and IFN signaling. The ubiquitination pathway and IFN signaling play crucial roles in the modification of antigen presentation and immune responses. Through KEGG enrichment analysis, we found that many proteins involved in antigen processing, peptidase function, ubiquitination pathways, and IFN signaling were altered after adjuvant treatment, particularly in the MPLA- and CpG-treated groups (Figure 5C; Figure S4B). The expression of each protein is shown in Figure S4C and Supplementary Table 2. These data suggest that MPLA and CpG adjuvants may affect the antigen processing of APCs, resulting in fewer peptide presentations.”

Recommendation 4. Lines 212-218: I think there needs to be more discussion of interpretation here. Only one of the low-stability peptides required low concentrations for CD4+ T cell responses in vitro. What about the other peptides in the analysis? Perhaps if the data is taken together there is not a clear pattern?

Thank you for the comment. In this study, epitope-specific CD4+ T-cells were expanded in vitro from the spleens of peptide-pool-immunized mice. T-cell responses to individual peptides were detected using ICS and FACS. Only one peptide, recA #23, with low binding stability, and one high-stability peptide, ureA #2, induced effective T-cell responses. Peptide ureA #3 with high stability induces low Th1 responses. The other peptides cannot induce CD4+ T-cell secreting IFN-γ (Data are shown in Author response image 2). Thus, we compared the strength of IFN-γ responses induced by these three peptides at a set of low concentrations. Data for other peptides without any response could not be taken together.

**Author response image 2. sa3fig2:** The expanded CD4+T cells from peptides immunized mice were screened for their response to the peptides in an ICS assay.

In this study, we used a peptide pool containing four low-stability peptides to vaccinate mice; however, only one peptide induced an effective CD4+ T-cell response. We speculate that the possible reasons are as follows. First, the number of peptides used for vaccination is too small. Only four low-stability peptides were synthesized and used to immunize mice. Three of these could not induce an effective T-cell response, possibly because of their low immunogenicity. If more peptides are synthesized and used, more peptides that induce T-cell responses may be observed. Second, epitope-specific T-cell responses are variable. Responses to the subdominant peptides can be inhibited by the dominant peptide. The subdominant peptide can become dominant by changing the peptide dose or in the absence of the dominant peptide. Thus, we believe that responses to the other three peptides may be detected if mice are immunized with a peptide pool that does not contain a response epitope.

The corresponding statements have been added to the Discussion section on page 13 lines 287–291 as “Unfortunately, only one peptide, recA #23, with low binding stability and induced significant Th1 responses, was identified in this study. To further confirm that low-stability peptides can induce stronger and higher TCR-affinity antigen-specific T-cell clonotype responses than high-stability peptides, further studies should monitor more peptides with different stabilities.”

Recommendation 5. There are some areas where additional editing to text would be beneficial due to grammar (eg lines 122-126; line 116, etc).

The manuscript has been edited by a professional language editing company.

**Reviewer #2 (Recommendations For The Authors):**

Recommendation 1. It is interesting that there was no difference in IFNg responses induced by different adjuvants.

Thank you for the comment. Possible reasons for the lack of difference in IFN-γ responses could be as follows. First, all adjuvants used in this study have been confirmed to effectively induce Th1 responses. Second, in this study, IFN-γ responses were examined using expanded antigen-specific T cells in vitro. The in vitro cell expansion efficiency may have affected these results.

Recommendation 2. The data to support the claim that changes in exogenous peptide presentation among adjuvant groups were not due to differences in antigen phagocytosis is insufficient.

Thank you for the comment. In this study, proteomics of A20 cells pulsed with antigens in different adjuvant-treated groups were used to determine exogenous antigens phagocytosed by cells. In addition, we used fluorescein isothiocyanate (FITC)-labeled OVA to pulse APCs and detected antigen phagocytosis by APCs after treatment with different adjuvants. The MFI of FITC was detected by FACS at different time points. The data are shown below (Author response image 3). No obvious differences in FITC MFI were detected after adjuvant stimulation, indicating that antigen phagocytosis among the adjuvant groups was almost the same.

A20 cells, used as APCs, are the B-cell line. Antigen recognition and phagocytosis by B-cells depends on the B-cell receptor (BCR) on the cell surface. The ability of BCRs to bind to different antigens varies, leading to significant differences in the phagocytosis of different antigens by B-cells. Therefore, detecting the phagocytosis of a single antigen may not reflect the overall phagocytic state of the B-cells. Thus, in this study, we used proteomics to detect exogenous proteins in B-cells pulsed with H. pylori antigens, which contain thousands of components, to evaluate their overall phagocytic capacity. Only the proteomic data are presented in our manuscript.

**Author response image 3. sa3fig3:** Antigen phagocytosis of A20 cells were measured using FITC-labeled OVA. (A) A20 cells were pulsed with FITC-labeled OVA. MFI of FITC was measured after 1 h. (B) MFI of FITC was examined post the stimulation of adjuvants at different time points.

Recommendation 3. It is not clear how MPLA, CpG, and MDP adjuvants modulate the presentation of low vs high stability peptides.

Thank you for pointing this out. We acknowledge that we did not clarify the mechanisms by which adjuvants affect the stability of the peptide presentations of APCs.

We performed experiments to detect the expression of proteins involved in antigen processing and presentation in the different adjuvant-treated groups. Furthermore, shRNAs were used to knock down the expression of key molecules. Immunopeptidomics was used to detect peptide presentation. Unfortunately, the expected results for peptide presentation repertoires were not observed. We are still working on the mechanisms.

Please also see our response to comment 1 of reviewer 1

The related statements were added in the Discussion section on page 13, lines 292–299: “In this study, we found that the peptide repertoires presented by APCs were significantly affected by the adjuvants CpG and MPLA, but not MDP. All three adjuvants belong to the PRR ligand adjuvant family. CpG and MPLA bind to TLRs and MDP is recognized by NOD2. Although the receptors are different, many common molecules are involved both in TLR and NLD pathway activation. Unfortunately, we did not demonstrate why the MDP had different impacts on peptide presentation compared with other adjuvants. Further investigation is required to clarify the mechanism by which MPLA, CpG, and MDP adjuvants modulate the presentation of peptides with different stabilities.”